ᵃ | **Open Peer Review** | Virology | Research Article

# Genome-wide bioinformatics analysis of human protease capacity for proteolytic cleavage of the SARS-CoV-2 spike glycoprotein

Evgenii V. Matveev,[1,2,3] Gennady V. Ponomarev,[1,2] Marat D. Kazanov[1,2,3,4]

**ABSTRACT** Severe acute respiratory syndrome coronavirus-2 (SARS-CoV-2) primarily enters the cell by binding the virus's spike (S) glycoprotein to the angiotensin-converting enzyme 2 receptor on the cell surface, followed by proteolytic cleavage by host proteases. Studies have identified furin and transmembrane protease serine 2 proteases in priming and triggering cleavages of the S glycoprotein, converting it into a fusion-competent form and initiating membrane fusion, respectively. Alternatively, SARS-CoV-2 can enter the cell through the endocytic pathway, where activation is triggered by lysosomal cathepsin L. However, other proteases are also suspected to be involved in both entry routes. In this study, we conducted a genome-wide bioinformatics analysis to explore the capacity of human proteases in hydrolyzing peptide bonds of the S glycoprotein. Predictive models of sequence specificity for 169 human proteases were constructed and applied to the S glycoprotein together with the method for predicting structural susceptibility to proteolysis of protein regions. After validating our approach on extensively studied S2′ and S1/S2 cleavage sites, we applied our method to each peptide bond of the S glycoprotein across all 169 proteases. Our results indicate that various members of the proprotein convertase subtilisin/kexin type, type II transmembrane family serine protease, and kallikrein families, as well as specific coagulation factors, are capable of cleaving S2′ or S1/S2 sites. We have also identified a potential cleavage site of cathepsin L at the K790 position within the S2′ loop. Structural analysis suggests that cleavage of this site induces conformational changes similar to the cleavage at the R815 (S2′) position, leading to the exposure of the fusion peptide and subsequent fusion with the membrane. Other potential cleavage sites and the influence of mutations in common SARS-CoV-2 variants on proteolytic efficiency are discussed.

**IMPORTANCE** The entry of severe acute respiratory syndrome coronavirus-2 (SARS-CoV-2) into the cell, activated by host proteases, is considerably more complex in coronaviruses than in most other viruses and is not fully understood. There is evidence that other proteases beyond the known furin and transmembrane protease serine 2 can activate the spike protein. Another example of uncertainty is the cleavage site for the alternative endocytic route of SARS-CoV-2 entrance, which is still unknown. Bioinformatics methods, modeling protease specificity and estimating the structural susceptibility of protein regions to proteolysis, can aid in studying this topic by predicting the involved proteases and their cleavage sites, thereby substantially reducing the amount of experimental work. Elucidating the mechanisms of spike protein activation is crucial for preventing possible future coronavirus pandemics and developing antiviral drugs.

**KEYWORDS** SARS-CoV-2, spike glycoprotein, proteolytic activation, protease, proteolysis, furin, TMPRSS2, cathepsin L, PCSK, TTSP

Address correspondence to Marat D. Kazanov, mkazanov@gmail.com.

The authors declare no conflict of interest.

See the funding table on p. 11.

Viruses typically hijack host cell proteins to perform essential tasks necessary for their life cycle, such as cell entry, replication, and assembly (1). Therefore, viruses generally lack genes in their genome that encode functions exploited from the host cellular machinery. Since specific functions are not encoded in viral genomes, it is challenging to identify which host factors are required for the life cycle of viruses. Cataloging host proteins involved in a viral life cycle is crucial for the development of antiviral drugs and therapeutic strategies (2). The experiments aimed at identifying these host factors are labor-intensive, making bioinformatics predictions beneficial for guiding experimental studies.

The COVID-19 pandemic underscored the importance of virology research and highlighted the need for a comprehensive understanding of the viral life cycle, including the identification of cellular host factors for severe acute respiratory syndrome coronavirus-2 (SARS-CoV-2) infection (3). SARS-CoV-2 is a positive-stranded RNA virus from the Coronaviridae family (4). SARS-CoV-2 is thought to involve several host factors (5), among which proteases play an important role (6). They play a key role in activating the fusion of the SARS-CoV-2 particle with the cell membrane and altering the conformation of the viral receptor, referred to as the spike (S) glycoprotein, which consists of the receptor-binding domain S1 and the fusion domain S2 (7–9).

Previous studies have identified two crucial cleavage sites on the SARS-CoV-2 S glycoprotein, processed by host proteases (10). The first cleavage site, known as S2′, is located in the S2 domain and becomes accessible after the S glycoprotein binds to the angiotensin-converting enzyme 2 (ACE2) cell receptor (11, 12). According to current understanding, this site undergoes cleavage by the membrane-anchored transmembrane protease serine 2 (TMPRSS2), initiating the process of coronavirus particle fusion with the cell membrane (13). The second one, designated as S1/S2, is located at the linking of the S1 and S2 domains and is thought to be cleaved by furin in Golgi organelles before virus packaging (14). This cleavage induces conformational changes in the S glycoprotein, transitioning it from a so-called "close" to "open" conformation that facilitates binding to the ACE2 receptor (15). This site is absent in closely related coronaviruses, and the sequence insertion at S1/S2 in SARS-CoV-2 has sparked discussions regarding its artificial or natural origin (16–19).

It has been demonstrated that, even in the absence of TMPRSS2, SARS-CoV-2 can still enter the cell through an alternative endocytic pathway called "the late pathway" (20, 21). Studies suggest that cathepsin L activates the fusion of the coronavirus particle with the endolysosomal membrane via cleavage around the S1/S2 site (22), while other studies propose cleavage near the S2′ site (23). This entrance pathway is still relatively less studied compared to the ACE2-mediated "early" pathway. Currently, it is the subject of ongoing research (24), particularly in light of recent reports indicating that the SARS-CoV-2 Omicron BA.1 variant strain shows increased dependence on the endocytic cell entry route (25).

Multiple studies showed that other proteases beyond furin, TMPRSS2, and cathepsin L may be involved in the proteolytic processing of the S glycoprotein (26). Thus, it has been demonstrated that coagulation factors, such as thrombin and factor Xa, can directly cleave the S glycoprotein (27). Another study reported the potential involvement of metalloproteinases in this proteolytic event (28). It was established earlier for other coronaviruses that members of type II transmembrane family serine proteases (TTSPs), including HAT and matriptase, can cleave the spike glycoprotein (10). The human genome contains more than 600 proteases, classified into five catalytic groups—aspartic, metallo, cysteine, serine, and threonine peptidases (29, 30), so this wide repertoire of proteases boosts the chance that some of them can be involved in activating S glycoprotein. Despite these individual cases, to our knowledge, there are no systematic studies of the role of other human proteases in SARS-CoV-2 infection. To fill this gap in the field, we conducted a genome-wide bioinformatics analysis to explore the potential involvement of human proteases in the cleavage of the S glycoprotein. This analysis has generated high-confidence predictions that can guide future experimental studies.

## MATERIALS AND METHODS

### Sequence and structural data, data on protease localization and expression

The amino acid sequence of the S glycoprotein from the Wuhan strain was obtained from UniProt (ID: P0DTC2) (31). The three-dimensional structures of the S glycoprotein in both the "open" [Protein Data Bank (PDB) ID: 6VYB] and "closed" (PDB ID: 6VXX) conformations, as well as its complex with the ACE2 receptor (PDB ID: 7DF4), were downloaded from the PDB (32). Loops were modeled using SWISS-MODEL with the default parameters (33). Data on SARS-CoV-2 mutations were obtained from the CoVariants database (34). Data on protease localization were taken from the COMPARTMENTS database (35). Protease expression in human tissues was obtained from the TISSUES database (36).

### Models of protease sequence specificity

Data on proteolytic events were retrieved from the MEROPS database (37). Models of protease sequence specificity were constructed in the form of positional-weighted matrices (PWMs) as described in reference (38). Briefly, for each human protease from the MEROPS database with at least eight proteolytic events, the frequencies of amino acid occurrences at P3–P3′ positions $f_{ij}$ were calculated [Schechter–Berger notation (39)], where $i$ represents the amino acid and $j$ represents the position. Background frequencies of amino acids $b_j$ were calculated for the whole human proteome. Each cell of the PWM, with columns representing P3–P3′ position rows representing 20 amino acids, was assigned a value $r_{ij}$ calculated as the logarithm of the ratio of the amino acid frequency at cleavage sites to the amino acid frequency at the background $r_{ij} = \log f_{ij}/b_j$. By considering the PWM positions as independent, the cleavage score (CS) for particular protein peptide bonds was calculated as the sum of the relevant values from each P3–P3′ column, corresponding to the amino acids observed at those positions. The P1 position is used as a reference point for specifying the positions of the cleavage sites throughout the paper.

### Model of the structural susceptibility to proteolysis

The method for estimation of the structural susceptibility to proteolysis was previously described in reference (40). Briefly, proteolytic events extracted from CutDB (41) were mapped to the corresponding three-dimensional structures of protease substrates. A set of predictors was composed from conventional structural features, such as solvent accessibility, temperature factor, secondary structure, and additional features like loop length and terminal regions. A training set was curated to exclude data from experiments with potentially denatured substrates, which were identified based on the presence of high numbers of cleavages in the hydrophobic protein core. The final model was trained using linear discriminant analysis, which demonstrated better performance compared to other conventional machine learning methods from the scikit-learn library (42).

## RESULTS

### Models of protease sequence specificity

To elucidate the capacity of human proteases in proteolytic processing of the SARS-CoV-2 spike glycoprotein, we initially constructed predictive models of sequence specificity for 169 human proteases (Supplemental File 1). We defined this subset of human proteases based on the availability of information on identified protease substrates taken from the MEROPS database (37). To construct predictive models of protease sequence specificity in the form of PWMs (38), we filtered the MEROPS data to identify human proteases with a sufficient number of substrates. We applied the obtained models to the amino acid sequence of the SARS-CoV-2 S glycoprotein from the original Wuhan strain to estimate the probability of cleavage for each protein peptide bond by individual proteases. To facilitate the interpretation of the obtained CSs, we first compared CS distributions for the known cleaved and uncleaved positions using

MEROPS data on known protease substrates (Fig. 1A). The results showed that proteolytic sites have Q1–Q3 CS values in the interval [1.64, 5.02], with mean and median values of 3.36 and 3.16, respectively, while uncleaved positions have Q1–Q3 CS values in the interval [−5.89, −1.03], with mean and median values of −3.57 and −3.42, respectively.

To validate our approach, we analyzed the calculated CSs for the S2′ (position 815) and S1/S2 (position 685) cleavage sites across a set of 169 considered human proteases (Fig. 1B and C). For the S2′ position, the highest scores were observed for TMPRSS2, which is consistent with the current understanding of the activation of the virus–cell membrane fusion process. Likewise, at the S1/S2 cleavage position, the highest CSs were obtained for furin, which aligns with the current knowledge of SARS-CoV-2 biology (43).

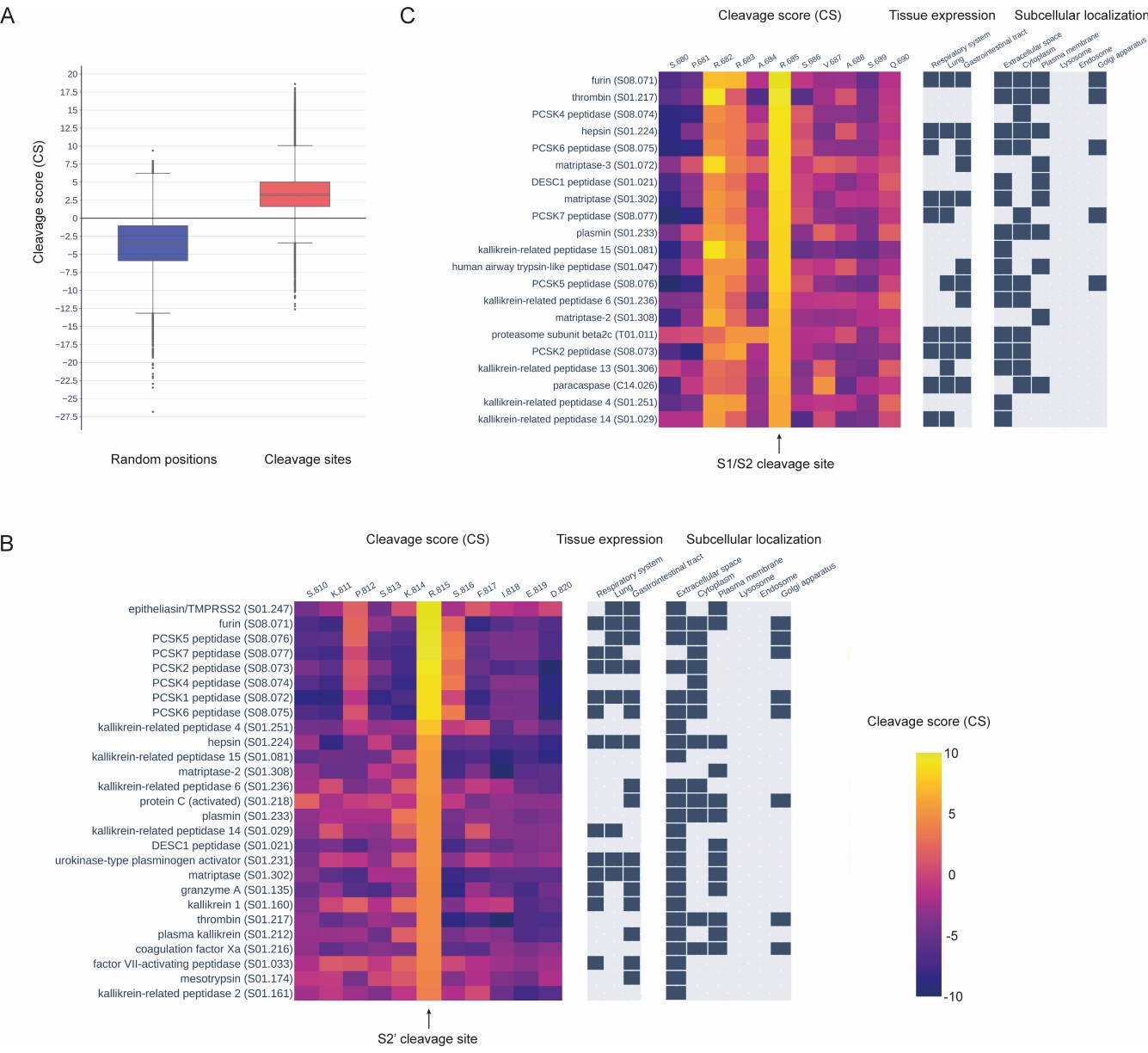

FIG 1 (A) A comparison of cleavage scores generated by protease sequence specificity models for known cleavage sites and random protein positions. (B) Cleavage scores, protease cellular localization, and protease tissue expression for the top 20 proteases with the highest cleavage scores at the S2′ (R815) position. (C) Cleavage scores, protease cellular localization, and protease tissue expression for the top 20 proteases with the highest cleavage scores at the S1/S2 (R685) position.

## S2′ cleavage site

Besides TMPRSS2, other members of the TTSPs (44)—hepsin, DESC1, matriptase, and matriptase-2—showed sufficiently high CS values at the S2′ position (Fig. 1B; Fig. S1A). Additionally, high scores for the S2′ site were obtained for furin and other members of the proprotein convertase subtilisin/kexin type (PCSK) family, including PCSK1, PCSK2, PCSK4, PCSK5, PCSK6, and PCSK7. The third protease family whose members showed sufficiently high CS values are the kallikreins. Thus, CS values were elevated for several members of this family—KLKB1, KLK1, KLK2, KLK4, KLK6, KLK14, and KLK15. A little lower but still significant CSs were observed for several coagulation factors.

To investigate the potential interactions between the mentioned proteases and the S glycoprotein during the coronavirus's life cycle inside a host cell, we analyzed data on protease localization (Fig. 1B; Fig. S1A). According to the data on the localization of proteins in human cells (see Materials and Methods), the above-discussed members of the TTSP family are localized in the plasma membrane, similar to the TMPRSS2 protease. Some members of the PCSK family, including furin, PCSK1, PCSK5, PCSK6, and PCSK7, are found in the Golgi apparatus, and all of them, excluding PCSK7 and including PCSK2, are localized in the extracellular space. The localization of the mentioned members of the kallikrein family is primarily in the extracellular space, while coagulation factors were found in the plasma membrane, extracellular space, and cytoplasm. Thus, many members of these protease families theoretically have the potential to proteolytically activate the process of SARS-CoV-2 fusion with the membrane of an infected cell.

We have also analyzed data on protease expression in human tissues (Fig. 1B; Fig. S1A). Available data point to the expression of three out of the five mentioned members of the TTSP family—TMPRSS2, hepsin, and matriptase—in the respiratory system, lungs, and gastrointestinal tract, i.e., three tissue types associated with SARS-CoV-2 spreading in the human body. Most of the aforementioned members of the PCSK family, including furin, PCSK1, PCSK2, PCSK5, PCSK6, and PCSK7, also have evidence of expression in at least one of these tissue types. Among specified kallikreins, only KLK14 has data indicating the expression in the respiratory system and lungs, while KLKB1 and KLK6 have data showing expression in the gastrointestinal tract. Urokinase is the only coagulation factor with expression data present in all three tissue types.

## S1/S2 cleavage site

We conducted the same analysis for the S1/S2 cleavage site as we did for the S2′ site (Fig. 1C; Fig. S1B). Alongside furin, which obtained the highest CS value at the S1/S2 position, we observed high cleavage scores for other members of the PCSK family, namely, PCSK2, PCSK4, PCSK5, PCSK6, and PCSK7. Additionally, we found sufficiently high CS values for coagulation factors such as thrombin and plasmin, as well as for hepsin, HAT, DESC1, and other members of the TTSP family. Slightly lower scores were observed for kallikreins. The known localization of HAT, a member of the TTSP family not described above, is in the extracellular space and plasma membrane. Data on tissue expression show that the HAT protease is present in the gastrointestinal tract, a tissue also linked to the spread of SARS-CoV-2.

## Cathepsin L cleavage site

Although the role of cathepsin L in activating the virus's entrance from the endolysosome to the cytoplasm is established for many coronaviruses (45), including SARS-CoV-2, the specific cathepsin L cleavage site is yet to be identified. There are contradictions in cleavage site positions according to the bioinformatics and experimental studies (46, 47). Our results for cathepsin L indicated that the K790 position has the highest CS value (Fig. 2A). This site is located close to the S2′, and we suggest that proteolytic cleavage of the K790 could result in similar conformational changes of the S glycoprotein as proteolysis at S2′ leading to the exposure of the fusion peptide, activating the coronavirus fusion with the membrane.

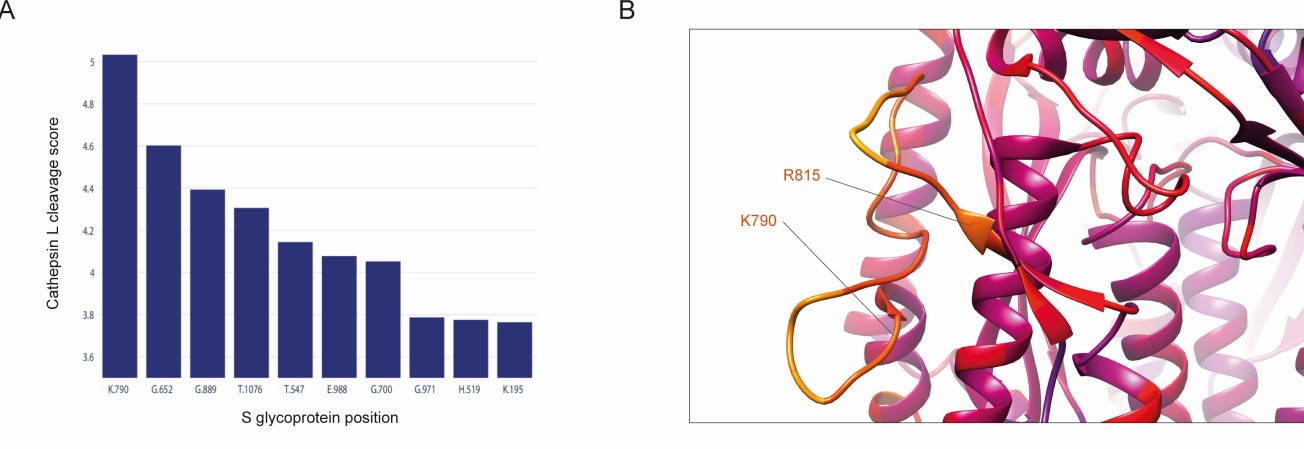

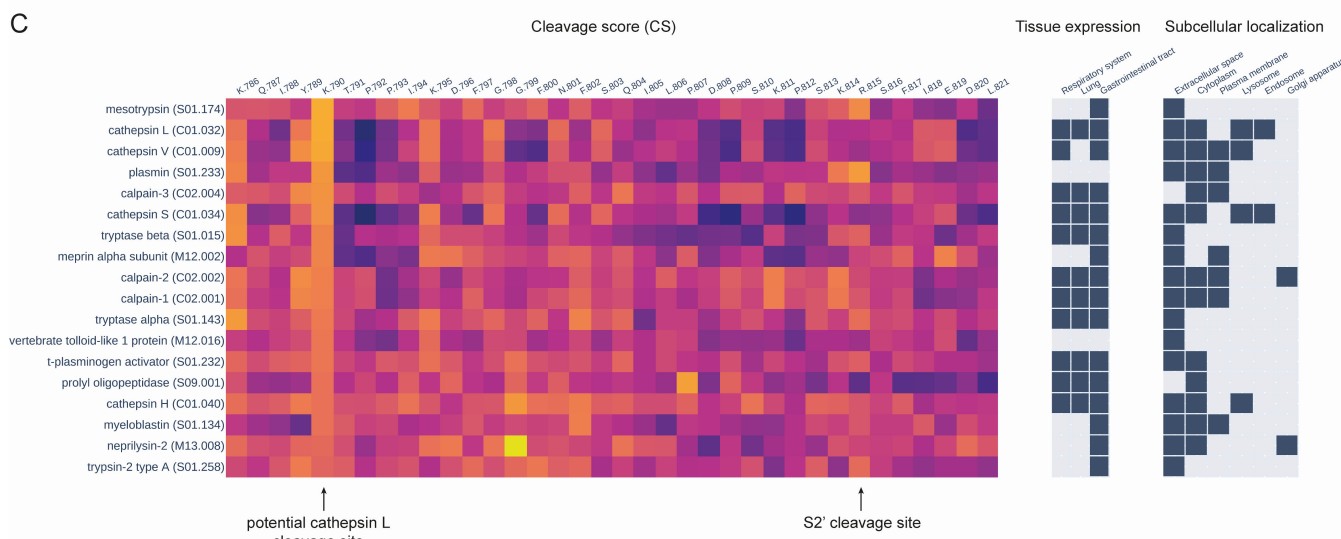

**FIG 2** (A) Top 10 positions of the S glycoprotein with the highest cleavage scores, generated by the sequence specificity model of cathepsin L. (B) The location of the presumable cathepsin L cleavage site K790 on the 3D structure of the SARS-CoV-2 spike glycoprotein. The K790 cleavage site is located within the S2′ loop, in close proximity to the known TMPRSS2 cleavage site R815. It is known that cleavage at the S2′ site leads to the exposure of the fusion peptide and initiates the fusion process between the coronavirus particle and the cell membrane. (C) Cleavage scores, protease cellular localization, and protease tissue expression for the top 20 proteases with the highest cleavage scores at the K790 position.

To assess the accessibility of K790 to proteolysis from the structural perspective, we applied to the 3D structure of the S glycoprotein a bioinformatics method, previously developed by our group, for predicting structural susceptibility to proteolysis of the protein regions (40). We mapped the calculated structural cleavage scores (SCSs) onto the 3D structure of the S glycoprotein in complex with the ACE2 receptor (Fig. 2B) and analyzed them specifically for the R815 and K790 sites. Both sites are located within the loop, which is adjacent to the fusion peptide. We observed that the SCSs were sufficiently high and quite comparable at both the K790 and R815 sites, with values of 0.61 and 0.58, respectively. Thus, we speculate that the K790 position could be a cleavage site of cathepsin L, whose proteolytic processing leads to the activation of fusion with the lysosomal membrane during the endocytic route of SARS-CoV-2 entry. We have also analyzed CSs calculated for other proteases at this position (Fig. 2C; Fig. S2). Interestingly, cathepsin L had the second-highest CS value, while the highest CS value was observed for mesotrypsin. The third-highest CS was obtained for another member of the papain family—cathepsin V. However, mesotrypsin is expressed solely in the gastrointestinal

tract, whereas cathepsin L's expression data indicate its presence across the respiratory system, lungs, and gastrointestinal tract.

## Other potential cleavage sites

To investigate other potential cleavage sites, we further analyzed the entire set of CSs obtained for the S glycoprotein sequence, limiting our analysis to the four mentioned protease families—PCSK, TTSP, kallikreins, and coagulation factors. We calculated the mean CS values across these four protease families for each S glycoprotein position and analyzed the top 20 positions with the highest CSs, together with the SCS values (Fig. 3A). Besides the known cleavage sites R685 and R815, only five positions among these 20 showed high values for both sequence and structural scores: R78, R346, R466, R634, and R847. The R78 site is located in the N-terminal domain (Fig. 3B), whereas R346 and R466 sites (Fig. 3C) are positioned within the receptor-binding domain (RBD). The R634 site is located within the loop connecting the S1 and S2 domains (Fig. 4A), known as the 630 loop (7). Contrary to the R815 site, which is located immediately upstream of the fusion peptide (FP), the R847 site is situated downstream of the FP (Fig. 4B). We then analyzed

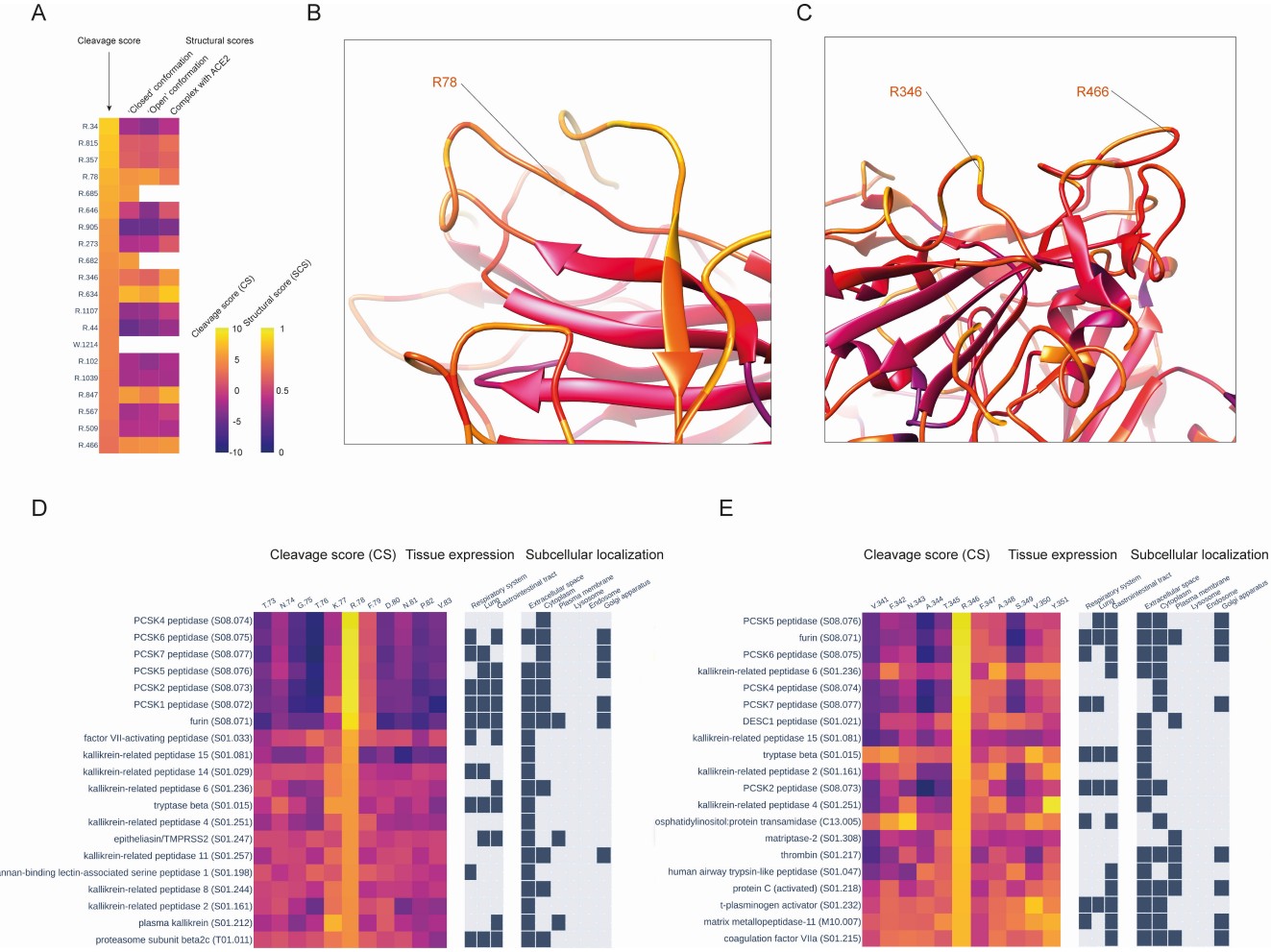

**FIG 3** (A) Top 20 positions of S glycoprotein with the highest mean cleavage score values across PCSK, TTSP, and kallikreins protease families. For each position, the scores of the structural susceptibilities to proteolysis were also calculated using 3D structures of the S glycoprotein in both "closed" and "open" conformations, as well as in the complex with the ACE2 receptor. (B) The location of the presumable cleavage site R78 within the N-terminal domain (NTD) of the SARS-CoV-2 spike glycoprotein. (C) The location of the presumable cleavage sites R346 and R466 within the RBD of the SARS-CoV-2 spike glycoprotein. (D) Cleavage scores, protease cellular localization, and protease tissue expression for the top 20 proteases with the highest cleavage scores at the R78 position. (E) Cleavage scores, protease cellular localization, and protease tissue expression for the top 20 proteases with the highest cleavage scores at the R346 position.

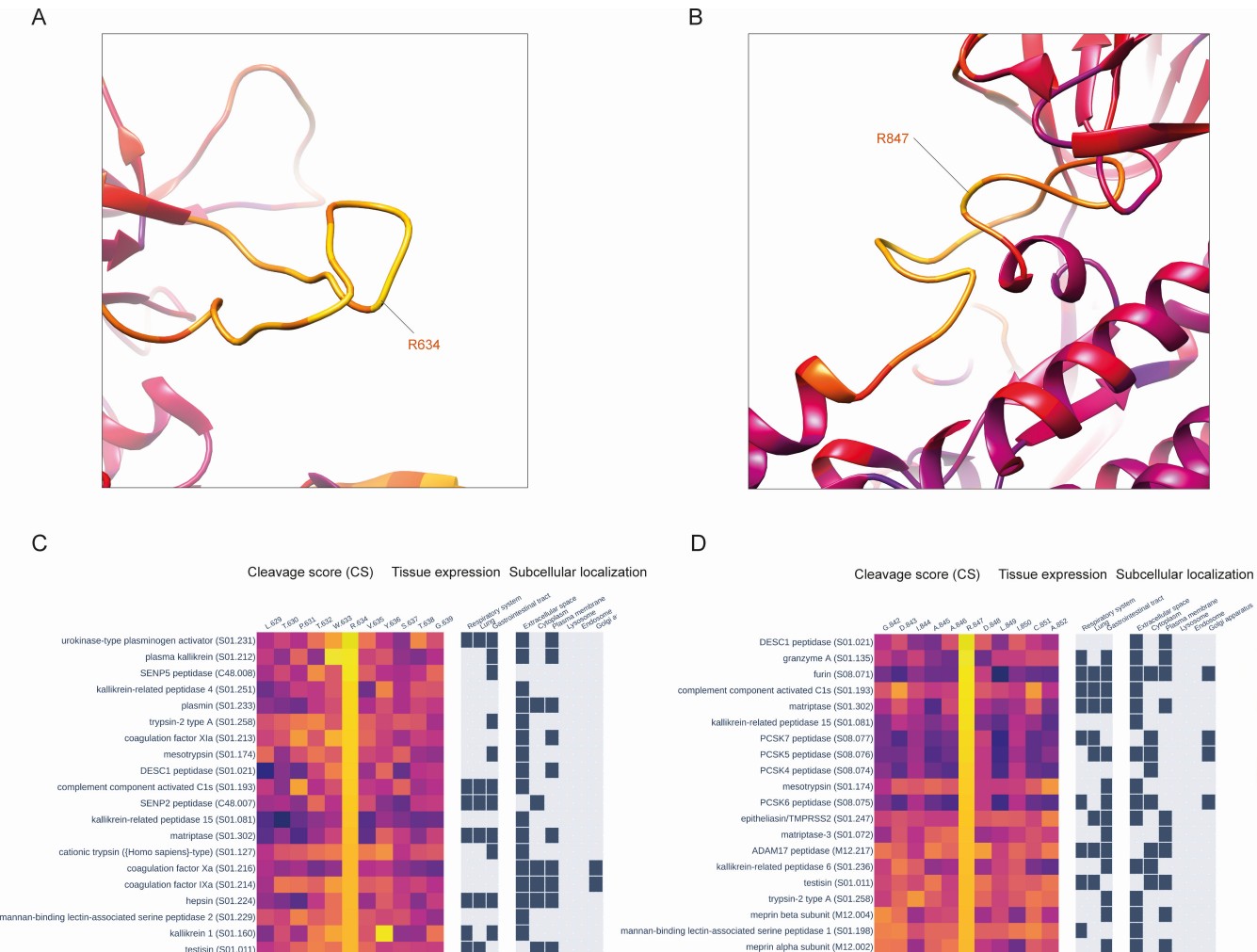

**FIG 4** (A) The location of the presumable cleavage site R634 within the so-called 630 loop of the SARS-CoV-2 spike glycoprotein. (B) The location of the presumable cleavage site R847 on the 3D structure of the SARS-CoV-2 spike glycoprotein. (C) Cleavage scores, protease cellular localization, and protease tissue expression for the top 20 proteases with the highest cleavage scores at the R634 position. (D) Cleavage scores, protease cellular localization, and protease tissue expression for the top 20 proteases with the highest cleavage scores at the R847 position.

cleavage scores at these positions for all 169 proteases. The cleavage scores at the R78 site were most favorable for proteases from the PCSK family, with all of them being approximately equal and significantly higher than scores calculated for other proteases (Fig. 3D; Fig. S3A). For the R346 site, the best CSs were obtained for the PCSK family, with PCSK5 peptidase and furin being the first and second top proteases, respectively (Fig. 3E; Fig. S3B). Cathepsin C, also known as dipeptidyl peptidase 1, showed the highest CS at the R466 site, followed by members of the PCSK family (Fig. S4A). Interestingly, for the R634 site, the best four CS values were obtained for proteases from three different classes (Fig. 4C; Fig. S4B)—serine (urokinase and plasma kallikrein), cysteine (SENP5), and glutamate (RCE1 peptidase). The highest CS at the R847 site was obtained for a member of the TTSP family—DESC peptidase (Fig. 4D; Fig. S5).

To gain insights into the difference between the cleavage scores of presumed cleavage sites and known cleavage sites S2′ and S1/S2, we created a heatmap visualizing the cleavage scores of all proteolytic sites mentioned in this study, calculated for proteases from PCSK, TTSP, kallikreins, and coagulation factor families (Fig. 5A). Analysis of the heatmap reveals the distinct preferences of each protease for specific cleavage sites. As anticipated, TMPRSS2 exhibited the highest cleavage score at the R815 site,

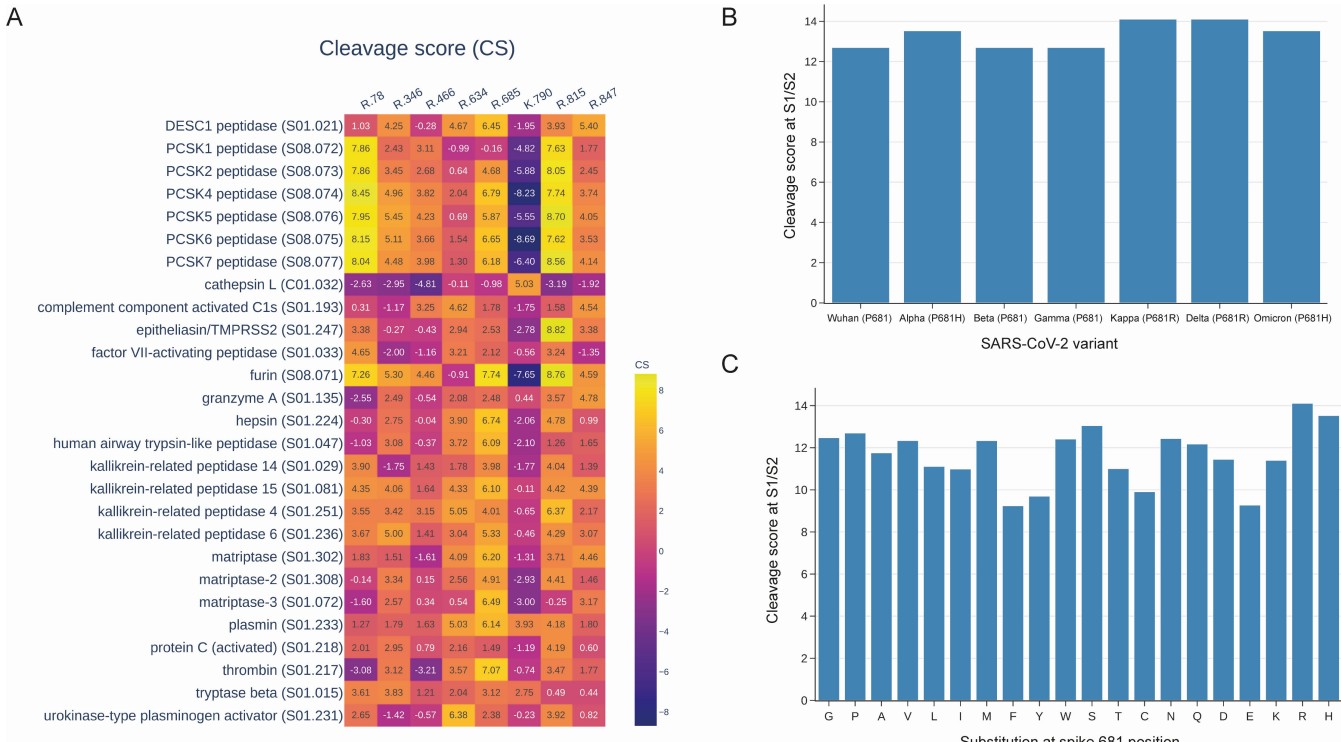

**FIG 5** (A) Cleavage scores computed for both known (R815, R685) and potential proteolytic sites using protease specificity models specifically developed for members of the PCSK, TTSP, kallikreins, and coagulation factor protease families. (B) Cleavage scores for the spike S1/S2 site, calculated using the P5–P5′ PWM, among diverse SARS-CoV-2 variants, including those carrying H/R substitutions at the P681 position. (C) Complete landscape of spike S1/S2 site cleavage scores across substitutions at the 681 position.

while for furin, the optimal sites are R685 and also R815. Interestingly, other members of the PCSK family have the top site at R78, while R685 is only the second best. This observation could possibly explain why other members of the PCSK family beyond furin were not reported to activate the S-glycoprotein.

## Influence of SARS-CoV-2 mutations on cleavage sites

We further analyzed whether any known SARS-CoV-2 mutations influenced the considered cleavage sites. Our analysis of mutations in positions around the cleavage sites was limited to three amino acids from both sides of the cleavage sites, in accordance with the size of the PWM. We found that out of the eight aforementioned P1 positions (78, 346, 466, 634, 685, 790, 815, and 847) and the three surrounding amino acids on each side of the cleaved peptide bond, only R346 was substituted by lysine and tyrosine in different SARS-CoV-2 variants. These independent substitutions significantly reduce CSs at this position (Fig. S6) and may indicate a selection toward the degradation of this presumable cleavage site.

We also paid special attention to the well-known mutations P681H/R, which are five amino acids away from the S1/S2 cleavage site. To obtain the computational estimates of the influence of these mutations on S1/S2 cleavage efficiency, we constructed a new, larger matrix of furin sequence specificity (PWM) by considering positions within five residues around the cleavage site. We applied a new furin PWM to S-glycoprotein's S1/S2 site of SARS-CoV-2 variants and found that Alpha and Omicron variants with the P681H mutation showed higher cleavage scores than the original Wuhan variant. Additionally, Kappa and Delta variants, carrying the P681R mutations, demonstrate even higher cleavage score values (Fig. 5B). We have also calculated a complete landscape of furin cleavage scores at the S1/S2 site relative to all possible substitutions at the

681 position (Fig. 5C). Interestingly, R and H substitutions were found to be associated with the highest and second-highest cleavage scores, respectively. Proline, the wild-type amino acid at position 681 in the S glycoprotein of the Wuhan strain, obtained the fourth-highest cleavage score, following serine. These results align with recent experiments (48–50).

To prevent biases in our structural estimation of susceptibility to proteolysis at the studied cleavage sites, associated with mutations in different SARS-CoV-2 variants and those introduced experimentally to enhance protein crystallizability, we included five additional structures related to various SARS-CoV-2 variants (Alpha, Beta, Delta, Kappa, and Omicron) in the analysis for each of the three S-glycoprotein conformations. We found that our structural estimates are sufficiently stable for all mentioned cleavage sites (Fig. S7).

## DISCUSSION

In this study, we presented a genome-wide bioinformatics analysis of the capabilities of human proteases to proteolytically process the spike glycoprotein of SARS-CoV-2. This analysis was conducted using protease sequence specificity models, which were built based on known protease substrates, along with the method for estimating the structural susceptibility to proteolysis developed previously by our group. We validated our approach on well-studied S1/S2 and S2′ cleavage sites of the S glycoprotein: our method identified furin and TMPRSS2 as the proteases with the highest cleavage scores for the S1/S2 and S2′ sites, respectively, and estimated that these sites are accessible for proteolysis in a structural sense. These results strongly align with the current knowledge of SARS-CoV-2 biology. It is worth noting that our structural method can consider the 3D structures of proteins with attached glycans. Although structures with glycans are underrepresented in the PDB due to challenges in producing and crystallizing glycosylated proteins (51), they can serve as input for our method. This can be important, as it is known that 40% of the surface of the spike protein is shielded by glycans (52, 53).

By applying this approach to every peptide bond of the S glycoprotein for all 169 human proteases, we thoroughly explored the proteolytic capacity of the subset of human proteolytic enzymes. For the known S1/S2 and S2′ cleavage sites, we found that, in addition to furin and TMPRSS2, there are other members of four families of serine proteases—PCSK, TTSP, kallikreins, and coagulation factors—which demonstrate theoretical potential for cleaving S1/S2 and S2′ sites based on their sequence specificity, colocalization, and expression in tissues associated with SARS-CoV-2 spread in the human body.

Our analysis focused on cathepsin L revealed a potential cleavage site at position K790, which is located close to S2′ and on the same loop. Subsequent structural examination showed the accessibility of this position for proteolysis and allowed us to suggest that cleavage of this site can induce structural changes similar to the cleavage of the S2′ R815 position, which leads to the exposure of the fusion peptide and activation of fusion with the membrane. Interestingly, our proposed cleavage site differs from the cathepsin L cleavage sites previously suggested by both bioinformatics and experimental studies. However, the robust validation results of our approach on known cleavage sites have allowed us to designate the K790 site as a strong candidate for experimental validation.

In addition to analyzing established cleavage sites, such as S2′ and S1/S2, for susceptibility to other proteases, and examining known implicated proteases like cathepsin L for their specific cleavage sites, we performed an unbiased search to identify additional potential cleavage sites of the S glycoprotein and the corresponding proteases capable of cleaving them. The search for additional cleavage sites was conducted in our study since we cannot exclude the possibility of unknown proteolytic processing of the spike glycoprotein in the SARS-CoV-2 life cycle. By analyzing both calculated sequence and structural proteolytic scores, we identified five positions on the S glycoprotein that likely correspond to potential cleavage sites: R78, R346, R466, R634,

and R847. These sites are located in various segments of the S glycoprotein including the NTD, RBD, the loop connecting the S1 and S2 regions, and downstream of the fusion peptide. We have also examined the known and potential cleavage sites for preferences by furin, TMPRSS2, and other proteases from the PCSK, TTSP, kallikreins, and coagulation factor families. Interestingly, while furin displays the highest cleavage scores at the S1/S2 and S2′ sites, other members of the PCSK family demonstrate greater cleavage score values at position R78 than at the S1/S2 site. We speculate that this may be a contributing factor explaining why these proteases with similar specificity to furin are not linked to the pre-activation of the S glycoprotein at the S1/S2 site.

We have also examined whether mutations found in common SARS-CoV-2 variants are located near the considered cleavage sites and have the potential to impact their proteolytic efficiency. However, since we chose a sufficiently narrow width of the PWM involving three amino acids on each side of the cleavage, we were only able to estimate the impact of mutations located in the immediate vicinity of the cleavage sites. This limited size of the PWM was chosen to increase the number of proteases with modeled specificity, as a smaller PWM size requires fewer identified substrates to build the PWM. Finally, we identified only two independent mutations that influenced the aforementioned cleavage sites, and both of them led to the degradation of the presumable R346 cleavage site.

To the best of our knowledge, our study represents the first systematic analysis of human protease capabilities in the proteolytic processing of the SARS-CoV-2 spike glycoprotein. Our bioinformatics predictions provide direction for further experimental exploration and validation. The comprehensive understanding of the role of host factors, including human proteases, in SARS-CoV-2 infection, will significantly contribute to the development of new therapeutic agents, effective treatment strategies, and the prevention of future pandemics.

## ACKNOWLEDGMENTS

We thank Irina Ponomareva for the design of the pre-print's layout.

The reported study was funded by the Russian Science Foundation, grant number 22-14-00132.

## AUTHOR AFFILIATIONS

[1]Center for Molecular and Cellular Biology, Skolkovo Institute of Science and Technology, Moscow, Russia

[2]Research and Training Center on Bioinformatics, A.A.Kharkevich Institute for Information Transmission Problems, Moscow, Russia

[3]Laboratory of Cytogenetics and Molecular Genetics, Dmitry Rogachev National Medical Research Center of Pediatric Hematology, Oncology and Immunology, Moscow, Russia

[4]Faculty of Engineering and Natural Sciences, Sabanci University, Istanbul, Turkey

## AUTHOR ORCIDs

Evgenii V. Matveev http://orcid.org/0009-0008-2103-4978
Gennady V. Ponomarev http://orcid.org/0000-0003-1271-9007
Marat D. Kazanov http://orcid.org/0000-0002-2314-5507

## FUNDING

| Funder | Grant(s) | Author(s) |
| --- | --- | --- |
| Russian Science Foundation (RSF) | 22-14-00132 | Marat D. Kazanov |

## AUTHOR CONTRIBUTIONS

Evgenii V. Matveev, Data curation, Formal analysis, Investigation, Resources, Software, Validation, Visualization | Gennady V. Ponomarev, Data curation, Formal analysis, Investigation | Marat D. Kazanov, Conceptualization, Formal analysis, Methodology, Project administration, Supervision, Writing – original draft, Writing – review and editing

## ADDITIONAL FILES

The following material is available online.

### Supplemental Material

**File S1 (Spectrum03530-23-s0001.xlsx).** Additional data.
**Supplemental Figures (Spectrum03530-23-s0002.pdf).** Figures S1 to S7.

### Open Peer Review

**PEER REVIEW HISTORY (review-history.pdf).** An accounting of the reviewer comments and feedback.

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
