## [Reviewer comments · Microbiology Spectrum]

Microbiology Spectrum

Genome-wide Bioinformatics Analysis of Human Protease Capacity for Proteolytic Cleavage of the SARS-CoV-2 Spike Glycoprotein

Eugenii Matveev, Gennady Ponomarev, and Marat Kazanov

Corresponding Author(s): Marat Kazanov, Sabanci Universitesi Muhendislik ve Doga Bilimleri Fakultesi

Review Timeline:

Submission Date:	October 3, 2023
Editorial Decision:	November 3, 2023
Revision Received:	December 6, 2023
Accepted:	December 7, 2023

Editor: Takamasa Ueno

Reviewer(s): The reviewers have opted to remain anonymous.

Transaction Report:

DOI: <https://doi.org/10.1128/spectrum.03530-23>

Re: Spectrum03530-23 (Genome-wide Bioinformatics Analysis of Human Protease Capacity for Proteolytic Cleavage of the SARS-CoV-2 Spike Glycoprotein)

Dear Dr. Marat Kazanov:

Thank you for submitting your work to Microbiology Spectrum. We have received the comments from the two experts in this field and both reviewers are largely positive to this study. Below you will find my comments, instructions from the Spectrum editorial office, and the reviewer comments.

Revision Guidelines

Sincerely,
Takamasa Ueno
Editor
Microbiology Spectrum

Reviewer #1 (Comments for the Author):

This study revealed the characteristics of the S protein of SARS-CoV-2 as a substrate of protease by genome-wide bioinformatics analysis using information from various databases and in silico analysis methods. The comprehensive analysis of the characteristics of S proteins (e.g., optimal cleavage sites) as protease substrates (cleavage sites) for 169 human cell proteases and the presentation of systematic information are significant. In addition to the amino acid sequence (structure and

chemical properties) of the substrate, the steric structure and flexibility of the substrate molecule (e.g., around the cleavage site) are important conditions for protease recognition of the substrate, and the authors have attempted certain analytical approaches in this regard using the methods they have developed. On the other hand, subcellular localization of proteases is also very important as a condition for protease recognition of substrates, and the COMPARTMENTS database was used, but this approach was not sufficient in this analysis.

Limitations of this study should be described. For example, if the structural characterization of S proteins as substrates is to be based on the data available in the PDB, the structures often have mutations that stabilize the structure or have fixed in certain conditions, and this should be taken into account. The known substrate information by protease is also likely to be a mixture of analyses under various different conditions, and a certain bias is expected for the amino acid information at the cleavage site.

The recognition (cleavage) site of cathepsin L is not yet clear, and it is significant that this study showed the candidate position of the cleavage sites by systematic analysis. On the other hand, the significance of searching for cleavage sites other than S1/S2 and S2' is unclear. The significance must be clarified.

SARS-CoV-2 has repeatedly gone through epidemic waves with the emergence of various mutant strains (variants). Mutations have accumulated in the S protein, and in particular, mutations in the S1/S2 sites have been identified in many significant variants. This point needs to be mentioned. The significance of this study will be enhanced if the effects of these S1/S2 site mutations on cleavage by various proteases can be systematically demonstrated. Especially for the Omicron variant, there have been conflicting reports in several papers in prestigious journals regarding the cleavability of the S protein and the availability of TMPRSS2, and it would be useful to present data from a different perspective to address these arguments.

Reviewer #2 (Comments for the Author):

The manuscript by Matveev et al. describes bioinformatics analysis of the capacities of human proteases for hydrolyzing the spike glycoprotein of SARS-CoV-2. The potential cleavage sites of the spike glycoprotein by each of 169 human proteases were predicted based on protease sequence specificity and the structural susceptibility of the substrate to proteolysis. The results showed that several human proteases have the potential to hydrolyze the spike glycoprotein at previously known cleavage sites (S2' and S1/S2) or their neighboring sites, possibly leading to the exposure of a fusion peptide to initiate virus membrane fusion. In addition, five previously unrecognized cleavage sites were identified. This study provides valuable information about proteolysis of spike glycoprotein of SARS-CoV-2 by human proteases, and could direct further experimental validation. I only have minor points for the authors to consider.

1. Please give a brief description about the total number (~600?) and type of human proteases in the introduction section.
2. It is known that the proteolytic cleavage of SARS-CoV-2 spike glycoprotein is modulated by glycosylation, and the glycosylation sites of the spike glycoprotein have been identified [Zhang et al., 2021, PNAS 118(47):e2109905118; Gong et al., 2021, Signal Transduct Target Ther 6(1):396]. The glycosylation needs to be taken into consideration when estimation of the structural susceptibility to proteolysis.
3. Some human proteases could cleave the spike glycoprotein at multiple sites. For instance, at least five sites with high cleavage scores were observed for furin (Figs. 1B, 1C, 3D, 3E, and 4D). Which one is preferred by furin? Given that the spike glycoprotein would convert into a fusion-competent form after the initial proteolytic cleavage, is it possible that the fusion-competent form would be further cleaved by the same or other protease at another site? Would multiple cleavage of the spike glycoprotein affect virus membrane fusion? These points need to be discussed in the discussion section.

Response to Reviewers

Dear Editors,

We thank reviewers for careful analysis of our manuscript and for useful suggestions and comments. The changes made in response to these comments are listed below.

Reviewer #1

SARS-CoV-2 has repeatedly gone through epidemic waves with the emergence of various mutant strains (variants). Mutations have accumulated in the S protein, and in particular, mutations in the S1/S2 sites have been identified in many significant variants. This point needs to be mentioned. The significance of this study will be enhanced if the effects of these S1/S2 site mutations on cleavage by various proteases can be systematically demonstrated. Especially for the Omicron variant, there have been conflicting reports in several papers in prestigious journals regarding the cleavability of the S protein and the availability of TMPRSS2, and it would be useful to present data from a different perspective to address these arguments.

In accordance with the reviewer's suggestion, we expanded our analysis of mutations around the cleavage sites analyzed in our study. Although, as we mentioned in the section "Influence of SARS-CoV-2 mutations on cleavage sites", the size of the positional weight matrix (PWM) used for modeling protease specificity was 6 (i.e., 3 amino acids from each side of the cleavage site), and most of the known SARS-CoV-2 mutations did not fall into the PWM frame, we agree that some distant amino acid variations in the sequence can affect the efficiency of proteolytic cleavage. Thus, one of the known mutations are the H/R mutations at the P681 position, which is 5 amino acids away from the S1/S2 site. To obtain computational estimates of the influence of these mutations on S1/S2 cleavage efficiency, we constructed a larger matrix of furin specificity, extending by 5 positions around the cleavage site. We then applied this matrix to estimate the effects of mutations at the P681 site for various SARS-CoV-2 variants. Additionally, we have evaluated the entire cleavage score landscape at the S1/S2 site for furin, considering all possible amino acid variations at the 681 position. The calculated cleavage scores are presented in Figures 5b and 5c, respectively, and the analysis of the results has been added to the section "Influence of SARS-CoV-2 mutations on cleavage sites".

Limitations of this study should be described. For example, if the structural characterization of S proteins as substrates is to be based on the data available in the PDB, the structures often have mutations that stabilize the structure or have fixed in certain conditions, and this should be taken into account. The known substrate information by protease is also likely to be a mixture of analyses under various different conditions, and a certain bias is expected for the amino acid information at the cleavage site.

To address potential biases suggested by the reviewer, we extended our structural estimation of susceptibility to proteolysis using other available structures of the S-glycoprotein in the PDB database. We included 5 additional structures for each of the three known conformations of the S-glycoprotein associated with different SARS-CoV-2 variants (Alpha, Beta, Delta, Kappa, Omicron). Thus, a total of 15 structures of the S-glycoprotein were additionally included in the analysis. We calculated structural estimates of susceptibility to proteolysis for all these structures and presented the scores calculated for protein regions containing cleavage sites mentioned in our study in Figures S11. We have also added a position from the hydrophobic core of the protein as a control, which is certainly unavailable for proteases. The discussion of the results has been added to the “Influence of SARS-CoV-2 mutations on cleavage sites” section.

The recognition (cleavage) site of cathepsin L is not yet clear, and it is significant that this study showed the candidate position of the cleavage sites by systematic analysis. On the other hand, the significance of searching for cleavage sites other than S1/S2 and S2' is unclear. The significance must be clarified.

In our study, we conducted a search for additional cleavage sites in spike glycoprotein, as we cannot exclude the possibility of unknown proteolytic processing in the SARS-CoV-2 life cycle. Additionally, we identified preferences of specific proteases for both known and presumable cleavage sites in the S-glycoprotein. Thus, as observed in Figure 5a, other potential cleavage sites have sufficiently high cleavage scores, and in the case of mutations degrading existing cleavage sites, these alternative sites may become more preferable for proteases, influencing the fate of the S-glycoprotein. We have added these considerations to the Discussion section.

Reviewer #2

Some human proteases could cleave the spike glycoprotein at multiple sites. For instance, at least five sites with high cleavage scores were observed for furin (Figs. 1B, 1C, 3D, 3E, and 4D). Which one is preferred by furin? Given that the spike glycoprotein would convert into a fusion-competent form after the initial proteolytic cleavage, is it possible that the fusion-competent form would be further cleaved by the same or other protease at another site? Would multiple cleavage of the spike glycoprotein affect virus membrane fusion? These points need to be discussed in the discussion section.

We thank the reviewer for raising this fair point. To answer this question, we composed a separate heatmap with the cleavage scores at multiple sites for furin and other proteases from the protease families proposed in cleavage of S-glycoprotein (Figure 5a). In this heatmap, the preference of a particular protease between the sites mentioned in the study can be observed. If multiple sites have high scores, the theoretical order in which the protease can cleave the sites becomes apparent. However, the question whether the multiple sites can be sequentially cleaved depends on the particular conformational changes induced by each cut. Unfortunately, it is quite challenging to model conformational changes that occur after each cleavage and thus to predict the next possible cleavages.

It is known that the proteolytic cleavage of SARS-CoV-2 spike glycoprotein is modulated by glycosylation, and the glycosylation sites of the spike glycoprotein have been identified [Zhang et al., 2021, PNAS 118(47):e2109905118; Gong et al., 2021, Signal Transduct Target Ther 6(1):396]. The glycosylation needs to be taken into consideration when estimation of the structural susceptibility to proteolysis.

We thank the reviewer for pointing out this very interesting topic. Our structural method is capable of considering attached glycans if they are present in the 3D structure since our method consider all atoms in the structure. However, structures with glycans are underrepresented in the PDB due to difficulties in producing and crystallizing proteins with glycosylation.

Another possibility to estimate the impact of glycosylation is to include its influence in the protease sequence specificity model. It is theoretically possible to extend the PWM model with additional weights in each position of the matrix, reflecting the impact of possible glycosylation. However, this impact still cannot be accurately quantified. For example, as can be seen from Figure 5c, mutations to histidine and arginine at the 681 position of the S glycoprotein increase the proteolytic efficiency in comparison with proline. At the same time, as shown in Zhang et al., as mentioned by the reviewer, these mutations decrease O-glycosylation in this region, which also increases proteolytic efficiency. Thus, Zhang et al. measured only the collective impact of the mutations and the reduction of O-glycosylation. To comprehensively understand the relative impact of glycosylation and mutations, further research is required.

Please give a brief description about the total number (~600?) and type of human proteases in the introduction section.

We have included the following sentence in the Introduction:

“The human genome contains more than 600 proteases, classified into five catalytic groups - aspartic, metallo, cysteine, serine, and threonine peptidases [29,30], so this wide repertoire of proteases boosts the chance that some of them can be involved in activating S glycoprotein.”

Sincerely,
Marat D. Kazanov, Ph.D. (on behalf of all authors)

Re: Spectrum03530-23R1 (Genome-wide Bioinformatics Analysis of Human Protease Capacity for Proteolytic Cleavage of the SARS-CoV-2 Spike Glycoprotein)

Dear Dr. Marat Kazanov:

Your manuscript has been accepted, and I am forwarding it to the ASM production staff for publication. Your paper will first be checked to make sure all elements meet the technical requirements. ASM staff will contact you if anything needs to be revised before copyediting and production can begin. Otherwise, you will be notified when your proofs are ready to be viewed.

Sincerely,
Takamasa Ueno
Editor
Microbiology Spectrum